# Benchmarking Robustness in Object Detection: Autonomous Driving when Winter is Coming

## Abstract

The ability to detect objects regardless of image corruptions or weather conditions is crucial for real-world applications of deep learning like autonomous driving. We here provide an easy-to-use benchmark to assess how object detection models perform when image quality degrades. The three resulting benchmark datasets, termed PASCAL-C, COCO-C and Cityscapes-C, contain a large variety of image corruptions. We show that a range of standard object detection models suffer a severe performance loss on corrupted images (down to 30–60% of the original performance). Furthermore, we provide evidence that increased performance on those benchmarks translates into increased robustness towards real-world "natural" distortions such as real-world fog, rain and snow. Using our benchmark we show that corruption robustness scales with performance on clean data and that a simple data augmentation trick—stylizing the training images—leads to a substantial increase in robustness for both synthetic and natural corruptions on all dataset. We envision our comprehensive benchmark to track future progress towards building robust object detection models. Benchmark, code and data are available at `https://....`

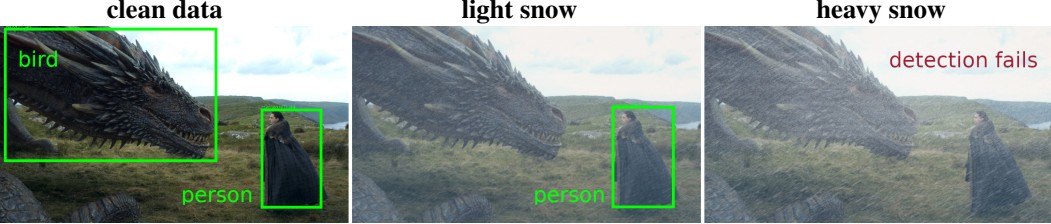

Figure 1: Mistaking a dragon for a bird (left) may be dangerous but missing it altogether because of snow (right) means playing with fire. Sadly, this is exactly the fate that an autonomous agent relying on a state-of-the-art object detection system would suffer. Predictions generated using Faster R-CNN; best viewed on screen.

## 1 Introduction

*A day in the near future: Autonomous vehicles are swarming the streets all over the world, tirelessly collecting data. But on this cold November afternoon traffic comes to an abrupt halt as it suddenly begins to snow: winter is coming. Huge snowflakes are falling from the sky and the cameras of autonomous vehicles are no longer able to make sense of their surroundings, triggering immediate emergency brakes. A day later, an investigation of this traffic disaster reveals that the unexpectedly large size of the snowflakes was the cause of the chaos: While state-of-the-art vision systems had been trained on a variety of common weather types, their training data contained hardly any snowflakes of this size...*

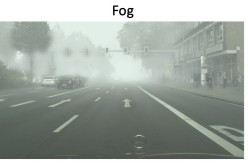 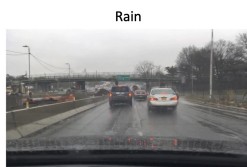 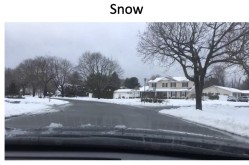 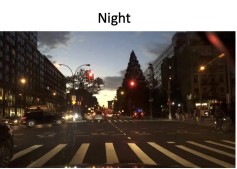

Figure 2: Expect the unexpected: To ensure safety, an autonomous vehicle must be able to recognize objects even in challenging outdoor conditions such as fog, rain, snow and at night.[1]

This fictional example highlights the problems that arise when Convolutional Neural Networks (CNNs) encounter settings that were not explicitly part of their training regime. For example, state-of-the-art object detection algorithms such as Faster R-CNN (Ren et al., 2015) fail to recognize objects when snow is added to an image (as shown in Figure 1), even though the objects are still clearly visible to a human eye. At the same time, augmenting the training data with several types of distortions is not a sufficient solution to achieve general robustness against previously unknown corruptions: It has recently been demonstrated that CNNs generalize poorly to novel distortion types, despite being trained on a variety of other distortions (Geirhos et al., 2018).

On a more general level, CNNs often fail to generalize outside of the training domain or training data distribution. Examples include the failure to generalize to images with uncommon poses of objects (Alcorn et al., 2019) or to cope with small distributional changes (e.g. Zech et al., 2018; Touvron et al., 2019). One of the most extreme cases are adversarial examples (Szegedy et al., 2013): images with a domain shift so small that it is imperceptible for humans yet sufficient to fool a DNN. We here focus on the less extreme but far more common problem of perceptible image distortions like blurry images, noise or natural distortions like snow.

As an example, autonomous vehicles need to be able to cope with wildly varying outdoor conditions such as fog, frost, snow, sand storms, or falling leaves, just to name a few (as visualized in Figure 2). One of the major reasons why autonomous cars have not yet gone mainstream is the inability of their recognition models to function well in adverse weather conditions (Dai & Van Gool, 2018). Getting data for unusual weather conditions is hard and while many common environmental conditions can (and have been) modelled, including fog (Sakaridis et al., 2018b), rain (Hospach et al., 2016), snow (Bernuth et al., 2019) and daytime to nighttime transitions (Dai & Van Gool, 2018), it is impossible to foresee all potential conditions that might occur "in the wild".

If we could build models that are robust to every possible image corruption, is is to be expected that weather changes would not be an issue. However, in order to assess the robustness of models one first needs to define a measure. While testing models on the set of all possible corruption types is impossible. We therefore propose to evaluate models on a diverse range of corruption types that were not part of the training data and demonstrate that this is a useful approximation for predicting performance under natural distortions like rain, snow, fog or the transition between day and night.

More specifically we propose three easy-to-use benchmark datasets termed PASCAL-C, COCO-C and Cityscapes-C to assess distortion robustness in object detection. Each dataset contains versions of the original object detection dataset which are corrupted with 15 distortions, each spanning five levels of severity. This approach follows Hendrycks & Dietterich (2019), who introduced corrupted versions of commonly used *classification* datasets (ImageNet-C, CIFAR10-C) as standardized benchmarks. After evaluating standard object detection algorithms on these benchmark datasets, we show how a simple data augmentation technique—stylizing the training images—can strongly improve robustness across corruption type, severity and dataset.

## 1.1 CONTRIBUTIONS

Our contributions can be summarized as follows:

1. We demonstrate that a broad range of object detection and instance segmentation models suffer severe performance impairments on corrupted images.

---

[1]Outdoor hazards have been directly linked to increased mortality rates (Lystad & Brown, 2018).

2. To quantify this behaviour and to enable tracking future progress, we propose the `Robust Detection Benchmark`, consisting of three benchmark datasets termed PASCAL-C, COCO-C & Cityscapes-C.

3. We demonstrate that improved performance on this benchmark of synthetic corruptions corresponds to increased robustness towards real-world "natural" distortions like rain, snow and fog.

4. We use the benchmark to show that corruption robustness scales with performance on clean data and that a simple data augmentation technique—stylizing the training data—leads to large robustness improvements for all evaluated corruptions without any additional labelling costs or architectural changes.

5. We make our benchmark, corruption and stylization code openly available in an easy-to-use fashion:

   - Benchmark, [2] data and data analysis are available at `https://...`[3]
   - Our pip installable image corruption library is available at `https://...`
   - Code to stylize arbitrary datasets is provided at `https://...`

## 1.2 RELATED WORK

**Benchmarking corruption robustness**   Several studies investigate the vulnerability of CNNs to common corruptions. Dodge & Karam (2016) measure the performance of four state-of-the-art image recognition models on out-of-distribution data and show that CNNs are in particular vulnerable to blur and Gaussian noise. Geirhos et al. (2018) show that CNN performance drops much faster than human performance for the task of recognizing corrupted images when the perturbation level increases across a broad range of corruption types. Azulay & Weiss (2018) investigate the lack of invariance of several state-of-the-art CNNs to small translations. A benchmark to evaluate the robustness of recognition models against common corruptions was recently introduced by Hendrycks & Dietterich (2019).

**Improving corruption robustness**   One way to restore the performance drop on corrupted data is to preprocess the data in order to remove the corruption. Mukherjee et al. (2018) propose a DNN-based approach to restore image quality of rainy and foggy images. Bahnsen & Moeslund (2018) and Bahnsen et al. (2019) propose algorithms to remove rain from images as a preprocessing step and report a subsequent increase in recognition rate. A challenge for these approaches is that noise removal is currently specific to a certain distortion type and thus does not generalize to other types of distortions. Another line of work seeks to enhance the classifier performance by the means of data augmentation, i.e. by directly including corrupted data into the training. Vasiljevic et al. (2016) study the vulnerability of a classifier to blurred images and enhance the performance on blurred images by fine-tuning on them. Geirhos et al. (2018) examine the generalization between different corruption types and find that fine-tuning on one corruption type does not enhance performance on other corruption types. In a different study, Geirhos et al. (2019) train a recognition model on a stylized version of the ImageNet dataset (Russakovsky et al., 2015), reporting increased general robustness against different corruptions as a result of a stronger bias towards ignoring textures and focusing on object shape. Hendrycks & Dietterich (2019) report several methods leading to enhanced performance on their corruption benchmark: Histogram Equalization, Multiscale Networks, Adversarial Logit Pairing, Feature Aggregating and Larger Networks.

**Evaluating robustness to environmental changes in autonomous driving**   In recent years, weather conditions turned out to be a central limitation for state-of-the art autonomous driving systems (Sakaridis et al., 2018b; Volk et al., 2019; Dai & Van Gool, 2018; Chen et al., 2018; Lee et al., 2018). While many specific approaches like modelling weather conditions (Sakaridis et al., 2018b;a; Volk et al., 2019; Bernuth et al., 2019; Hospach et al., 2016; Bernuth et al., 2018) or collecting real (Wen et al., 2015; Yu et al., 2018; Che et al., 2019; Caesar et al., 2019) and artificial (Gaidon et al., 2016; Ros et al., 2016; Richter et al., 2017; Johnson-Roberson et al., 2017) datasets with varying weather conditions, no general solution towards the problem has yet emerged. Radecki

---

[2]Our evaluation code to assess performance under corruption has been integrated into one of the most widely used detection toolboxes (URL omitted to keep anonymity during review period).

[3]All URLs omitted to keep anonymity for double-blind reviewing

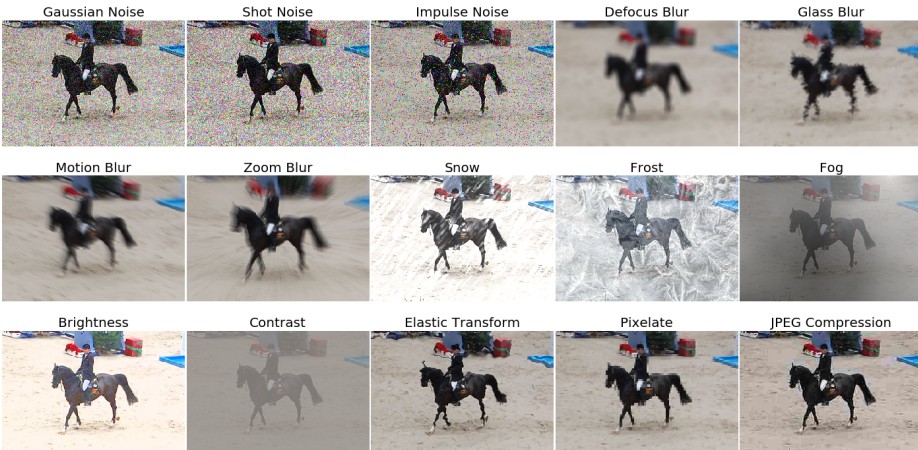

Figure 3: 15 corruption types from Hendrycks & Dietterich (2019), adapted to corrupt arbitrary images (example: randomly selected PASCAL VOC image, center crop, severity 3). Best viewed on screen.

et al. (2016) experimentally test the performance of various sensors and object recognition and classification models in adverse weather and lighting conditions. Bernuth et al. (2018) report a drop in the performance of a Recurrent Rolling Convolution network trained on the KITTI dataset when the camera images are modified by simulated raindrops on the windshield. Pei et al. (2017) introduce VeriVis, a framework to evaluate the security and robustness of different object recognition models using real-world image corruptions such as brightness, contrast, rotations, smoothing, blurring and others. Machiraju & Channappayya (2018) propose a metric to evaluate the degradation of object detection performance of an autonomous vehicle in several adverse weather conditions evaluated on the Virtual KITTI dataset. Building upon Hospach et al. (2016), Volk et al. (2019) study the fragility of an object detection model against rainy images, identify corner cases where the model fails and include images with synthetic rain variations into the training set. They report enhanced performance on real rain images. Bernuth et al. (2019) model photo-realistic snow and fog conditions to augment real and virtual video streams. They report a significant performance drop of an object detection model when evaluated on corrupted data.

## 2 METHODS

### 2.1 ROBUST DETECTION BENCHMARK

We introduce the `Robust Detection Benchmark` inspired by the ImageNet-C benchmark for object classification (Hendrycks & Dietterich, 2019) to assess object detection robustness on corrupted images.

**Corruption types**   Following Hendrycks & Dietterich (2019), we provide 15 corruptions on five severity levels each (visualized in Figure 3) to assess the effect of a broad range of different corruption types on object detection models.[4] The corruptions are sorted into four groups: noise, blur, digital and weather groups (as defined by Hendrycks & Dietterich (2019)). It is important to note that the corruption types are *not* meant to be used as a training data augmentation toolbox, but rather to measure a model's robustness against *previously unseen* corruptions. Thus, training should be done without using any of the provided corruptions. For model validation, four separate corruptions are provided (Speckle Noise, Gaussian Blur, Spatter, Saturate). The 15 corruptions described above should only be used to test the final model performance.

---

[4]These corruption types were introduced by Hendrycks & Dietterich (2019) and modified by us to work with images of arbitrary dimensions. Our generalized corruptions can be found at `https://...` and installed via `pip3 install ....`

**Benchmark datasets** The `Robust Detection Benchmark` consists of three benchmark datasets: PASCAL-C, COCO-C and Cityscapes-C. Among the vast number of available object detection datasets (Everingham et al., 2010; Geiger et al., 2012; Lin et al., 2014; Cordts et al., 2016; Zhou et al., 2017; Neuhold et al., 2017; Krasin et al., 2017), we chose to use PASCAL VOC (Everingham et al., 2010), MS COCO (Lin et al., 2014) and Cityscapes (Cordts et al., 2016) as they are the most commonly used datasets for general object detection (PASCAL & COCO) and street scenes (Cityscapes). We follow common conventions to select the tests splits: VOC2007 test set for PASCAL-C, the COCO 2017 validation set for COCO-C and the Cityscapes validation set for Cityscapes-C.

**Metrics** Since performance measures differ between the original datasets, the dataset-specific performance (P) measures are adopted as defined below:

$$P := \begin{cases} AP^{50}(\%) & \text{PASCAL VOC} \\ AP(\%) & \text{MS COCO } \& \text{ Cityscapes} \end{cases}$$

where $AP^{50}$ stands for the PASCAL 'Average Precision' metric at 50% Intersection over Union (IoU) and AP stands for the COCO 'Average Precision' metric which averages over IoUs between 50% and 95%. On the corrupted data, the benchmark performance is measured in terms of mean performance under corruption (mPC):

$$mPC = \frac{1}{N_c} \sum_{c=1}^{N_c} \frac{1}{N_s} \sum_{s=1}^{N_s} P_{c,s} \qquad (1)$$

Here, $P_{c,s}$ is the dataset-specific performance measure evaluated on test data corrupted with corruption $c$ under severity level $s$ while $N_c = 15$ and $N_s = 5$ indicate the number of corruptions and severity levels, respectively. In order to measure relative performance degradation under corruption, the relative performance under corruption (rPC) is introduced as defined below:

$$rPC = \frac{mPC}{P_{clean}} \qquad (2)$$

rPC measures the relative degradation of performance on corrupted data compared to clean data.

**Submissions** Submissions to the benchmark should be handed in as a simple pull request to the `Robust Detection Benchmark`[5] and need to include all three performance measures: clean performance ($P_{clean}$), mean performance under corruption (mPC) and relative performance under corruption (rPC). While mPC is the metric used to rank models on the `Robust Detection Benchmark`, the other measures provide additional insights, as they disentangle gains from higher clean performance (as measured by $P_{clean}$) and gains from better generalization performance to corrupted data (as measured by rPC).

**Baseline models** We provide baseline results for a set of common object detection models including Faster R-CNN (Ren et al., 2015), Mask R-CNN (He et al., 2017), Cascade R-CNN (Cai & Vasconcelos, 2018), Cascade Mask R-CNN (Chen et al., 2019a), RetinaNet (Lin et al., 2017b) and Hybrid Task Cascade (Chen et al., 2019a). We use a ResNet50 (He et al., 2016) with Feature Pyramid Networks (Lin et al., 2017a) as backbone for all models except for Faster R-CNN where we additionally test ResNet101 (He et al., 2016), ResNeXt101-32x4d (Xie et al., 2017) and ResNeXt-64x4d (Xie et al., 2017) backbones. We additionally provide results for Faster R-CNN and Mask R-CNN models with deformable convolutions (Dai et al., 2017; Zhu et al., 2018) in Appendix D. Models were evaluated using the `mmdetection toolbox` (Chen et al., 2019b); all models were trained and tested with standard hyperparameters. The details can be found in Appendix A.

## 2.2 STYLE TRANSFER AS DATA AUGMENTATION

For image classification, style transfer (Gatys et al., 2016)—the method of combining the content of an image with the style of another image—has been shown to strongly improve corruption robustness (Geirhos et al., 2019). We here transfer this method to object detection datasets testing two settings: (1) Replacing each training image with a stylized version and (2) adding a stylized version of each

---

[5]`https://...`

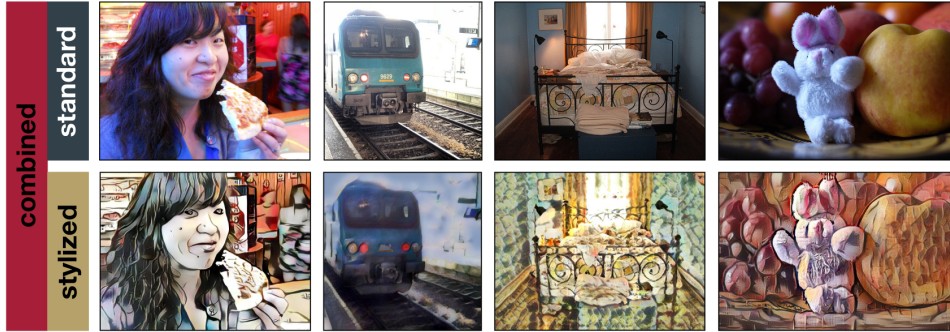

Figure 4: Training data visualization for COCO and Stylized-COCO. The three different training settings are: standard data (top row), stylized data (bottom row) and the concatenation of both (termed 'combined' in plots).

image to the existing dataset. We apply the fast style transfer method AdaIN (Huang & Belongie, 2017) with hyperparameter $\alpha = 1$ to the training data, replacing the original texture with the randomly chosen texture information of Kaggle's `Painter by Numbers`[6] dataset. Examples for the stylization of COCO images are given in Figure 4. We provide ready-to-use code for the stylization of arbitrary datasets at `https://...`

### 2.3 NATURAL DISTORTIONS

**Foggy Cityscapes**  Foggy Cityscapes Sakaridis et al. (2018b) is a version of Cityscapes with synthetic fog in three severity levels (given byt he attenuation coefficient $\beta = 0.005m^{-1}$, $0.01m^{-1}$ and $0.02m^{-1}$), that was carefully designed to look as realistic as possible. We use Fogy Cityscapes only at test time, testing the same models as used for our experiments with the original Cityscapes dataset and report results in the same AP metric.

**BDD100k**  BDD100k Yu et al. (2018) is a driving dataset consisting of 100 thousand videos of driving scenes recorded in varying conditions including weather changes and different times of the day[7]. We use these annotations to perform experiments, on different weather conditions ("clear", "rainy" and "snowy") and on the transition from day to night. Training is performed on what we would consider "clean" data - clear for weather and daytime for time - and evaluation is performed on all three splits. We use Faster R-CNN with the same hyper-parameters as in our experiments on COCO. Details of the dataset preparation can be found in Appendix C.

## 3 RESULTS

### 3.1 IMAGE CORRUPTIONS REDUCE MODEL PERFORMANCE

In order to assess the effect of image corruptions, we evaluated a set of common object detection models on the three benchmark datasets defined in Section 2. Performance is heavily degraded on corrupted images (compare Table 1). While Faster R-CNN can retain roughly 60% relative performance (rPC) on the rather simple images in PASCAL VOC, the same model suffers a dramatic reduction to 33% rPC on the Cityscapes dataset, which contains many small objects. With some variations, this effect is present in all tested models and also holds for instance segmentation tasks (for instance segmentation results, please see Appendix D).

### 3.2 ROBUSTNESS INCREASES WITH BACKBONE CAPACITY

We test variants of Faster R-CNN with different backbones (top of Table 1) and different head architectures (bottom of Table 1) on COCO. For the models with different backbones, we find that

---

[6]`https://www.kaggle.com/c/painter-by-numbers/`
[7]The frame at the 10th second of each video is annotated with additional information including bounding boxes which we use for our experiments

| PASCAL VOC | | | | |
|---|---|---|---|---|
| model | backbone | clean P [AP$^{50}$] | corrupted mPC [AP$^{50}$] | relative rPC [%] |
| Faster | r50 | 80.5 | 48.6 | 60.4 |

| MS COCO | | | | |
|---|---|---|---|---|
| model | backbone | clean P [AP] | corrupted mPC [AP] | relative rPC [%] |
| Faster | r50 | 36.3 | 18.2 | 50.2 |
| Faster | r101 | 38.5 | 20.9 | 54.2 |
| Faster | x101-32x4d | 40.1 | 22.3 | 55.5 |
| Faster | x101-64x4d | 41.3 | 23.4 | 56.6 |
| Mask | r50 | 37.3 | 18.7 | 50.1 |
| Cascade | r50 | 40.4 | 20.1 | 49.7 |
| Cascade Mask | r50 | 41.2 | 20.7 | 50.2 |
| RetinaNet | r50 | 35.6 | 17.8 | 50.1 |
| HTC | x101-64x4d | 50.6 | 32.7 | 64.7 |

| Cityscapes | | | | |
|---|---|---|---|---|
| model | backbone | clean P [AP] | corrupted mPC [AP] | relative rPC [%] |
| Faster | r50 | 36.4 | 12.2 | 33.4 |
| Mask | r50 | 37.5 | 11.7 | 31.1 |

Table 1: Object detection performance of various models. Backbones indicated with $r$ are ResNet and $x$ ResNeXt. All model names except for RetinaNet and HTC indicate the corresponding model from the R-CNN family. All COCO models were downloaded from the `mmdetection` modelzoo. For all reported quantities: higher is better; square brackets denote metric.

all image corruptions—except for the blur types—induce a fixed penalty to model performance, independent of the baseline performance on clean data: $\Delta \, \mathrm{mPC} \approx \Delta \, \mathrm{P}$ (compare Table 1 and Appendix Figure 10). Therefore, models with more powerful backbones show a relative performance improvement under corruption.[8] In comparison, Mask R-CNN, Cascade R-CNN and Cascade Mask R-CNN which draw their performance increase from more sophisticated head architectures all have roughly the same rPC of $\approx 50\%$. The current state-of-the-art model Hybrid Task Cascade (Chen et al., 2019a) is in so far an exception as it employs a combination of a stronger backbone, improved head architecture and additional training data to not only outperform the strongest baseline model by 9% AP on clean data but distances itself on corrupted data by a similar margin, achieving a leading relative performance under corruption (rPC) of 64.7%. These results indicate that robustness in the tested regime can be improved primarily through a better image encoding, and better head architectures cannot extract more information if the primary encoding is already sufficiently impaired.

### 3.3 TRAINING ON STYLIZED DATA IMPROVES ROBUSTNESS

In order to reduce the strong effect of corruptions on model performance observed above, we tested whether a simple approach (stylizing the training data) leads to a robustness improvement. We evaluate the exact same model (Faster R-CNN) with three different training data schemes (visualized in Figure 4):

**standard:** the unmodified training data of the respective dataset
**stylized:** the training data is stylized completely
**combined:** concatenation of standard and stylized training data

The results across our three datasets PASCAL-C, COCO-C and Cityscapes-C are visualized in Figure 5. We observe a similar pattern as reported by Geirhos et al. (2019) for object classification on ImageNet—a model trained on stylized data suffers less from corruptions than the model trained only on the original clean data. However, its performance on clean data is much lower. Combining stylized and clean data seems to achieve the best of both worlds: high performance on clean data as well as strongly improved performance under corruption. From the results in Table 2, it can be seen that both stylized and combined training improve the relative performance under corruption

---

[8]This finding is further supported by investigating models with deformable convolutions (see Appendix D).

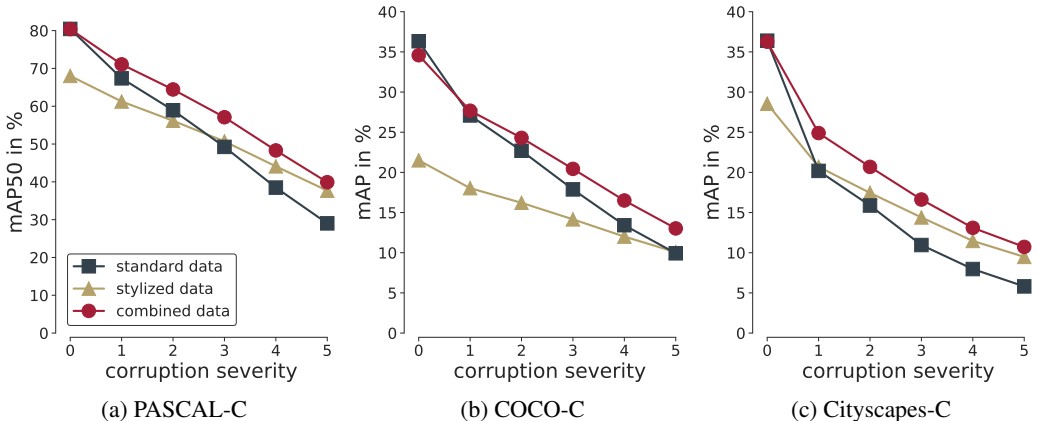

Figure 5: Figure 5: Training on stylized data improves test performance of Faster R-CNN on corrupted versions of PASCAL VOC, MS COCO and Cityscapes which include all 15 types of corruptions shown in Figure 3. Corruption severity 0 denotes clean data. Corruption specific performances are shown in the appendix (Figures 7, 8, 9).

| train data | PASCAL VOC [AP$^{50}$] | | | MS COCO [AP] | | | Cityscapes [AP] | | |
|---|---|---|---|---|---|---|---|---|---|
| | clean | corr. | rel. | clean | corr. | rel. | clean | corr. | rel. |
| | P | mPC | rPC [%] | P | mPC | rPC [%] | P | mPC | rPC [%] |
| standard | **80.5** | 48.6 | 60.4 | **36.3** | 18.2 | 50.2 | **36.4** | 12.2 | 33.4 |
| stylized | 68.0 | 50.0 | **73.5** | 21.5 | 14.1 | **65.6** | 28.5 | 14.7 | **51.5** |
| combined | 80.4 | **56.2** | 69.9 | 34.6 | **20.4** | 58.9 | 36.3 | **17.2** | 47.4 |

Table 2: Object detection performance of Faster R-CNN trained on standard images, stylized images and the combination of both evaluated on standard test sets (test 2007 for PASCAL VOC; val 2017 for MS COCO, val for Cityscapes); higher is better.

(rPC). Combined training yields the highest absolute performance under corruption (mPC) for all three datasets. This pattern is fairly consistent. Detailed results across corruption types are reported in the Appendix (Figure 7, Figure 8 and Figure 9).

### 3.4 TRAINING DIRECTLY ON STYLIZED DATA IS BETTER THAN USING STYLIZED DATA ONLY DURING PRE-TRAINING

For comparison reasons, we reimplemented the object detection models from Geirhos et al. (2019) and tested them for corruption robustness. Those models use backbones which are pre-trained with Stylized-ImageNet, but the object detection models are trained on the standard clean training sets of Pascal VOC and COCO. In contrast we here use backbones trained on standard "clean" ImageNet and train using stylized Pascal VOC and COCO. We find that stylized pre-training helps not only on clean data (as reported by Geirhos et al. (2019)) but also for corruption robustness (Table 3), albeit less than our approach of performing the final training on stylized data (compare to Table 2)[9].

### 3.5 ROBUSTNESS TO NATURAL DISTORTIONS IS CONNECTED TO SYNTHETIC CORRUPTION ROBUSTNESS

A central question is whether results on the robust detection benchmark generalize to real-world natural distortions like rain, snow or fog as illustrated in Figure 2. We test this using BDD100k (Yu et al., 2018), a driving scene dataset with annotations for weather conditions. For our first experiment, we train a model only on images that are taken in "clear" weather. We also train models on a stylized version of the same images as well as the combination of both following the protocol from Section 3.3. We then test these models on images which are annotated to be "clear", "rainy" or "snowy" (see

---

[9]Note that Geirhos et al. (2019) use Faster R-CNN without Feature Pyramids (FPN), which is why the baseline performance of these models is different from ours

| | PASCAL VOC [AP$^{50}$] | | | MS COCO [AP] | | |
|---|---|---|---|---|---|---|
| | clean | corr. | rel. | clean | corr. | rel. |
| train data | P | mPC | rPC [%] | P | mPC | rPC [%] |
| IN | 78.9 | 45.7 | 57.4 | 31.8 | 15.5 | 48.7 |
| SIN | 75.1 | 48.2 | 63.6 | 29.8 | 15.3 | 51.3 |
| SIN+IN | 78.0 | **50.6** | **64.2** | 31.1 | 16.0 | **51.4** |
| SIN+IN ft IN | **79.0** | 48.9 | 61.4 | **32.3** | **16.2** | 50.1 |

Table 3: Object detection performance of Faster R-CNN pre-trained on ImageNet (IN), Stylized ImageNet (SIN) and the combination of both evaluated on standard test sets (test 2007 for PASCAL VOC; val 2017 for MS COCO); higher is better.

| BDD100k [AP] | | Weather | | | | Day/Night | | |
|---|---|---|---|---|---|---|---|---|
| | clear | rainy | rel. | snowy | rel. | day | night | rel. |
| train data | P | mPC | rPC [%] | mPC | rPC [%] | P | mPC | rPC [%] |
| clean | **27.8** | 27.6 | 99.3 | 23.6 | 84.9 | **30.0** | 21.5 | 71.7 |
| stylized | 20.9 | 21.0 | 100.5 | 18.7 | **89.5** | 24.0 | 16.8 | 70.0 |
| combined | 27.7 | **28.0** | **101.1** | **24.2** | 87.4 | **30.0** | **22.5** | **75.0** |

Table 4: Performance of Faster R-CNN across different weather conditions and time changes when trained on standard images, stylized images and the combination of both evaluated on BDD100k (see Appendix C for dataset details); higher is better.

Appendix C for details). We find that these weather changes have little effect on performance on all three models, but that combined training improves the generalization to "rainy" and "snowy" images (Table 4 Weather). It may be important to note that the weather changes of this dataset are often relatively benign (e.g., images annotated as rainy often show only wet roads instead of rain).

A stronger test is generalization of a model trained on images taken during daytime to images taken at night which exhibit a strong appearance change. We find that a model trained on images taken during the day performs much worse at night but combined training improves nighttime performance (Table 4 Day/Night and Appendix C).

As a third test of real-world distortions, we test our approach on Foggy Cityscapes Sakaridis et al. (2018b) which uses fog in three different strengths (given by the attenuation factor $\beta = 0.005$, 0.01 or $0.2m^{-1}$) as a highly realistic model of natural fog. Fog drastically reduces the performance of standard models trained on Cityscapes which was collected in clear conditions. The reduction is almost 50% for the strongest corruption, see Table 5. In this strong test for OOD (out-of-distribution) robustness, stylized training increases relative performance substantially from about 50% to over 70% (Table 5).

Taken together, these results suggest that there is a connection between performance on synthetic and natural corruptions. Our approach of combined training with stylized data improves performance in every single case with increasing gains in harder conditions.

## 3.6 PERFORMANCE DEGRADATION DOES NOT SIMPLY SCALE WITH PERTURBATION SIZE

We investigated whether there is a direct relationship between the impact of a corruption on the pixel values of an image and the impact of a corruption on model performance. The left of Figure 6 shows the relative performance of Faster R-CNN on the corruptions in PASCAL-C dependent on the perturbation size of each corruption measured in Root Mean Square Error (RMSE). It can be seen that no simple relationship exists, counterintuitively robustness increases to corruption types with higher perturbation size (there is a weak positive correlation between rPC and RMSE, $r = 0.45$). This stems from the fact that corruptions like Fog or Brightness alter the image globally (resulting in high RMSE) while leaving local structure unchanged. Corruptions like Impulse Noise alter only a few pixels (resulting in low RMSE) but have a drastic impact on model performance.

To investigate further if classical perceptual image metrics are more predictive, we look at the relationship between the perceived image quality of the original and corrupted images measured in structural similarity (SSIM, higher value means more similar, Figure 6 on the right). There is a weak

| Foggy Cityscapes [AP] | $\beta = 0.005$ | | | $\beta = 0.01$ | | $\beta = 0.02$ | |
|---|---|---|---|---|---|---|---|
| | clean | corr. | rel. | corr. | rel. | corr. | rel. |
| train data | P | mPC | rPC [%] | mPC | rPC [%] | mPC | rPC [%] |
| standard | **36.4** | 30.2 | 83.0 | 25.1 | 69.0 | 18.7 | 51.4 |
| stylized | 28.5 | 26.2 | **91.9** | 24.7 | **86.7** | 22.5 | **78.9** |
| combined | 36.3 | **32.2** | 88.7 | **29.9** | 82.4 | **26.2** | 72.2 |

Table 5: Object detection performance of Faster R-CNN on Foggy Cityscapes when trained on Cityscapes with standard images, stylized images and the combination of both evaluated on the validation set; higher is better; $\beta$ is the attenuation coefficient in $m^{-1}$

correlation between rPC and SSIM ($r = 0.48$). This analysis shows that SSIM better captures the effect of the corruptions on model performance.

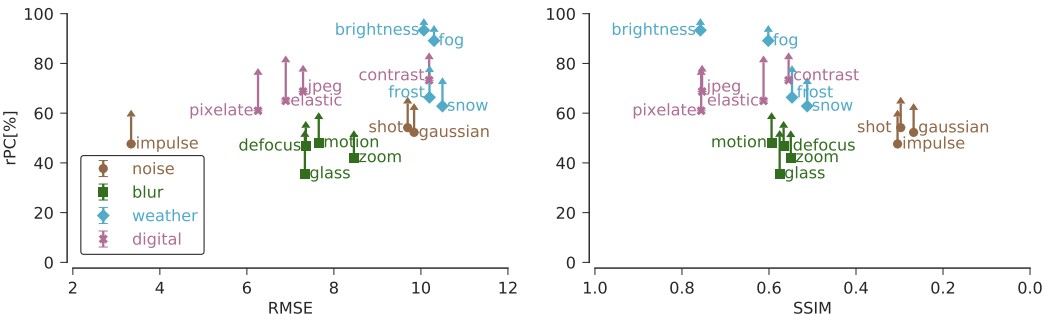

Figure 6: Relative performance under corruption (rPC) as a function of corruption RMSE (left, higher value=greater change in pixel space) and SSIM (right, higher value=higher perceived image quality) evaluated on PASCAL VOC. The dots indicate the rPC of Faster R-CNN trained on standard data; the arrows show the performance gained via training on 'combined' data. Corruptions are grouped into four corruption types: noise, blur, weather and digital.

## 4 DISCUSSION

We here showed that object detection and instance segmentation models suffer severe performance impairments on corrupted images. This drop in performance has previously been observed in image recognition models (e.g. Geirhos et al., 2018; Hendrycks & Dietterich, 2019). In order to track future progress on this important issue, we propose the `Robust Detection Benchmark` containing three easy-to-use benchmark datasets PASCAL-C, COCO-C and Cityscapes-C. We provide evidence that performance on our benchmarks predicts performance on natural distortions and show, that robustness corresponds to model performance on clean data. Apart from providing baselines, we demonstrate how a simple data augmentation technique, namely adding a stylized copy of the training data in order to reduce a model's focus on textural information, leads to strong robustness improvements. On corrupted images, we consistently observe a performance increase (about 16% for PASCAL, 12% for COCO, and 41% for Cityscapes) with small losses on clean data (0–2%). This approach has the benefit that it can be applied to any image dataset, requires no additional labelling or model tuning and, thus, comes basically for free. At the same time, our benchmark data shows that there is still space for improvement and it is yet to be determined whether the most promising robustness enhancement techniques will require architectural modifications, data augmentation schemes, modifications to the loss function, or a combination of these.

We encourage readers to expand the benchmark with novel corruption types. In order to achieve robust models, testing against a wide variety of different image corruptions is necessary—there is no 'too much'. Since our benchmark is open source, we welcome new corruption types and look forward to your pull requests to `https://...`! We envision our comprehensive benchmark to track future progress towards building robust object detection models that can be reliably deployed 'in the wild', eventually enabling them to cope with unexpected weather changes, corruptions of all kinds and, if necessary, even the occasional dragonfire.

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

## APPENDIX

### A  IMPLEMENTATION DETAILS: MODEL TRAINING

We train all our models with two images per GPU which corresponds to a batch size of 16 on eight GPUs. On COCO, we resize images so that their short edge is 800 pixels and train for twelve epochs with a starting learning rate of 0.01 which is decreased by a factor of ten after eight and eleven epochs. On PASCAL VOC, images are resized so that their short edge is 600 pixels. Training is done for twelve epochs with a starting learning rate of 0.00125 with a decay step of factor ten after nine epochs. For Cityscapes, we stayed as close as possible to the procedure described in (He et al., 2017), rescaling images to a shorter edge size between 800 and 1024 pixels and train for 64 epochs (to match 24k steps at a batch size of eight) with an initial learning rate of 0.0025 and a decay step of factor ten after 48 epochs. For evaluation, only one scale (1024 pixels) is used. Specifically, we used four GPUs to train the COCO models and one GPU for all other models[10] Training with stylized data is done by simply exchanging the dataset folder or adding it to the list of dataset folders to consider. For all further details please refer to the config files in our implementation (which we will make available after the end of the anonymous review period).

### B  CORRUPTING ARBITRARY IMAGES

In the original corruption benchmark of ImageNet-C (Hendrycks & Dietterich, 2019), two technical aspects are hard-coded: The image-dimensions and the number of channels. To allow for different data sets with different image dimensions, several corruption functions are defined independently of each other, such as `make_cifar_c`, `make_tinyimagenet_c`, `make_imagenet_c` and `make_imagenet_c_inception`. Additionally, many corruptions expect quadratic images. We have modified the code to resolve these constraints and now all corruptions can be applied to non-quadratic images with varying sizes, which is a necessary prerequisite for adapting the corruption benchmark to the PASCAL VOC and COCO datasets. For the corruption type Frost, crops from provided images of frost are added to the input images. Since images in PASCAL VOC and COCO have arbitrarily large dimensions, we resize the frost images to fit the largest input image dimension if necessary. The original corruption benchmark also expects RGB images. Our code now allows for grayscale images.[11] Both `motion_blur` and `snow` relied on the motion-blur functionality of Imagemagick, resulting in an external dependency that could not be resolved by standard Python package managers. For convenience, we reimplemented the motion-blur functionality in Python and removed the dependency on non-Python software.

### C  BDD100K

We use the weather annotations present in the BDD100k dataset Yu et al. (2018) to split it in images with clear, rainy and snowy conditions. We disregard all images which are annotated to have any other weather condition (foggy, partly cloudy, overcast and undefined) to make the separation easier[12]. We use all images from the training set which are labeled having clear weather conditions for training. For testing, we created 3 subsets of the validation set each containing 725 images in clear, rainy or snowy conditions[13]. The sets were created to have the same size which was determined by the category with the least images (rainy). Having same sized test sets is important because evaluation under the AP metric leads to lower scores with increasing sequence length Gupta et al. (2019).

---

[10]In all our experiments, we employ the linear scaling rule (Goyal et al., 2017) to select the appropriate learning rate.

[11]There are approximately 2–3% grayscale images in PASCAL VOC/MS COCO.

[12]It would have been great to combine the performance on natural fog with the results from Foggy Cityscapes but as there are only 13 foggy images in the validation set the results cannot be seen as representative in any way

[13]We will release the datasets splits at `https://...`

|  |  | MS COCO | | |
|---|---|---|---|---|
|  |  | clean | corr. | rel. |
| model | backbone | P [AP] | mPC [AP] | rPC [%] |
| Mask | r50 | 34.2 | 16.8 | 49.1 |
| Cascade Mask | r50 | 35.7 | 17.6 | 49.3 |
| HTC | x101-64x4d | 43.8 | 28.1 | 64.0 |

|  |  | Cityscapes | | |
|---|---|---|---|---|
|  |  | clean | corr. | rel. |
| model | backbone | P [AP] | mPC [AP] | rPC [%] |
| Mask | r50 | 32.7 | 10.0 | 30.5 |

Table 6: **Instance segmentation** performance of various models. Backbones indicated with $r$: ResNet. All model names indicate the corresponding model from the R-CNN family. All models were downloaded from the `mmdetection` modelzoo.

| | MS COCO | | | Cityscapes | | |
|---|---|---|---|---|---|---|
| | clean | corr. | rel. | clean | corr. | rel. |
| train data | [P] | [mPC] | [rPC] | [P] | [mPC] | [rPC] |
| standard | **34.2** | 16.9 | 49.4 | **32.7** | 10.0 | 30.5 |
| stylized | 20.5 | 13.2 | **64.1** | 23.0 | 11.3 | **49.2** |
| combined | 32.9 | **19.0** | 57.7 | 32.1 | **14.9** | 46.3 |

Table 7: **Instance segmentation** performance of Mask R-CNN trained on standard images, stylized images and the combination of both evaluated on standard test sets (test 2007 for PASCAL VOC; val 2017 for MS COCO, val for Cityscapes).

## D ADDITIONAL RESULTS

### D.1 INSTANCE SEGMENTATION RESULTS

We evaluated Mask R-CNN and Cascade Mask R-CNN on instance segmentation. The results are very similar to those on the object detection task with a slightly lower relative performance ( 1%, see Table 6). We also trained Mask R-CNN on the stylized datasets finding again very similar trends for the instance segmentation task as for the object detection task (Table 7). On the one hand, this is not very surprising as Mask R-CNN and Faster R-CNN are very similar. On the other hand, the contours of objects can change due to the stylization process, which would expectedly lead to poor segmentation performance when training only on stylized images. We do not see such an effect but rather find the instance segmentation performance of Mask R-CNN to mirror the object detection performance of Faster R-CNN when trained on stylized images.

### D.2 DEFORMABLE CONVOLUTIONAL NETWORKS

We tested the effect of deformable convolutions (Dai et al., 2017; Zhu et al., 2018) on corruption robustness. Deformable convolutions are a modification of the backbone architecture exchanging some standard convolutions with convolutions that have adaptive filters in the last stages of the encoder. It has been shown that deformable convolutions can help on a range of tasks like object detection and instance segmentation. This is the case here too as networks with deformable convolutions do not only perform better on clean but also on corrupted images improving relative performance by 6-7% compared to the baselines with standard backbones (See Tables 8 and 9). The effect appears to be the same as for other backbone modifications such as using deeper architectures (See Section 3 in the main paper).

### IMAGE RIGHTS & ATTRIBUTION

Figure 1: Home Box Office, Inc. (HBO).

MS COCO

| model | backbone | clean P [AP] | corr. mPC [AP] | rel. rPC [%] |
|---|---|---|---|---|
| Faster | r50-dcn | 40.0 | 22.4 | 56.1 |
| Faster | x101-64x4d-dcn | 43.4 | 26.7 | 61.6 |
| Mask | r50-dcn | 41.1 | 23.3 | 56.7 |

Table 8: **Object detection** performance of models with deformable convolutions Dai et al. (2017). Backbones indicated with r are ResNet, the addition dcn signifies deformable convolutions in stages c3-c5. All model names indicate the corresponding model from the R-CNN family. All models were downloaded from the `mmdetection` modelzoo.

MS COCO

| model | backbone | clean P [AP] | corr. mPC [AP] | rel. rPC [%] |
|---|---|---|---|---|
| Mask | r50-dcn | 37.2 | 20.7 | 55.7 |

Table 9: **Instance segmentation** performance of Mask R-CNN with deformable convolutions (Dai et al., 2017). The backbone indicated with $r$ is a ResNet 50, the addition dcn signifies deformable convolutions in stages c3-c5. The model was downloaded from the `mmdetection` modelzoo.

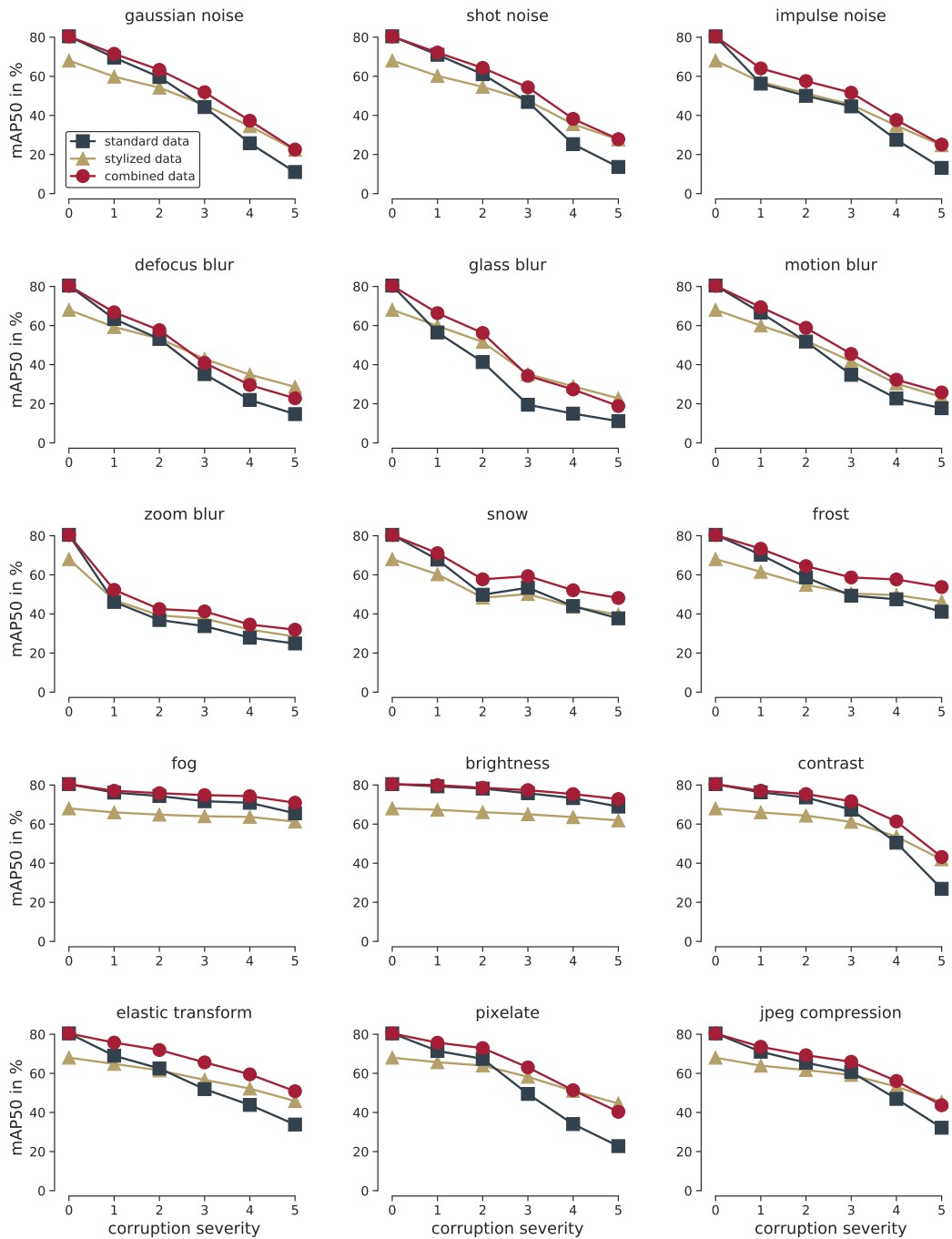

Figure 7: Results for each corruption type on PASCAL-C.

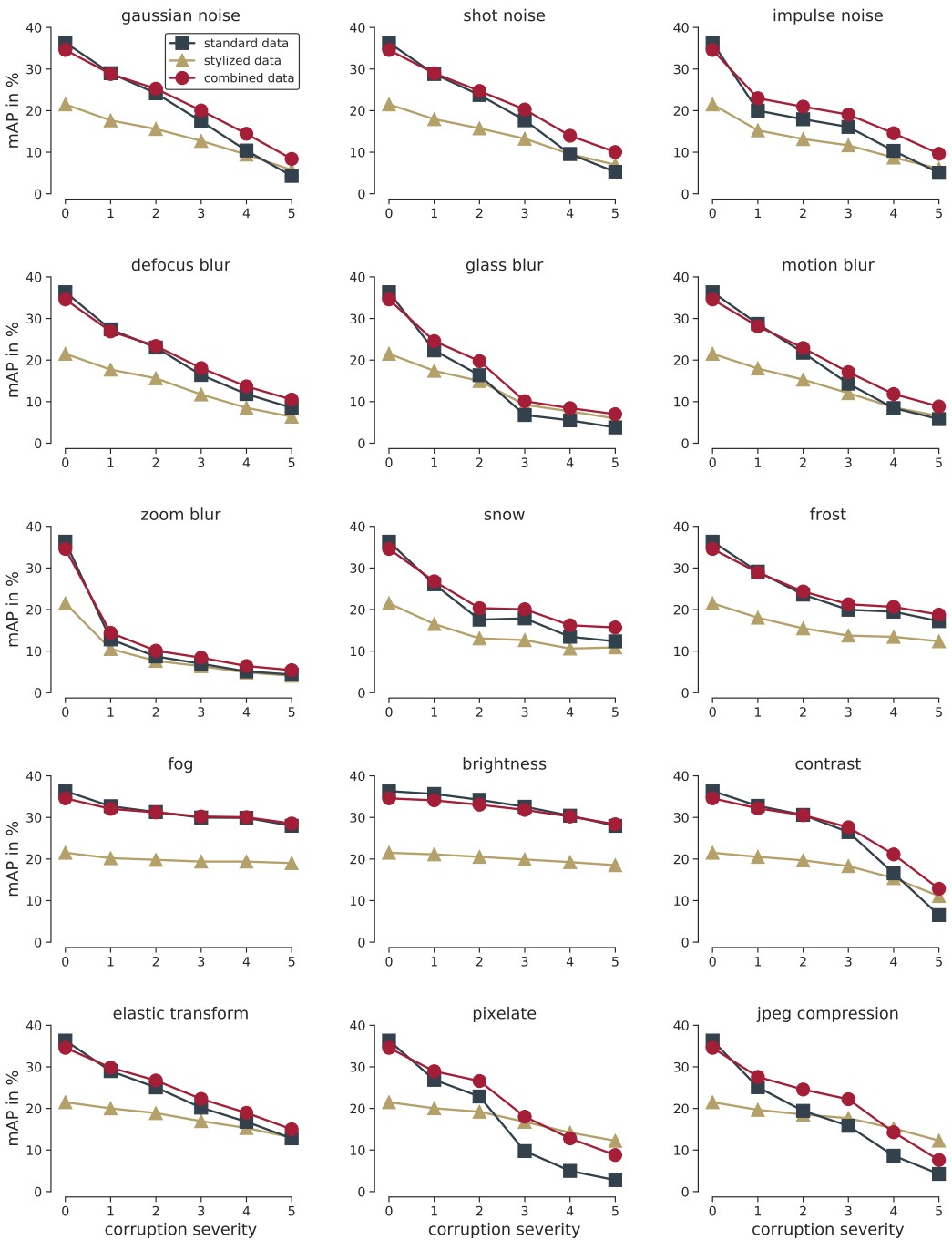

Figure 8: Results for each corruption type on COCO-C.

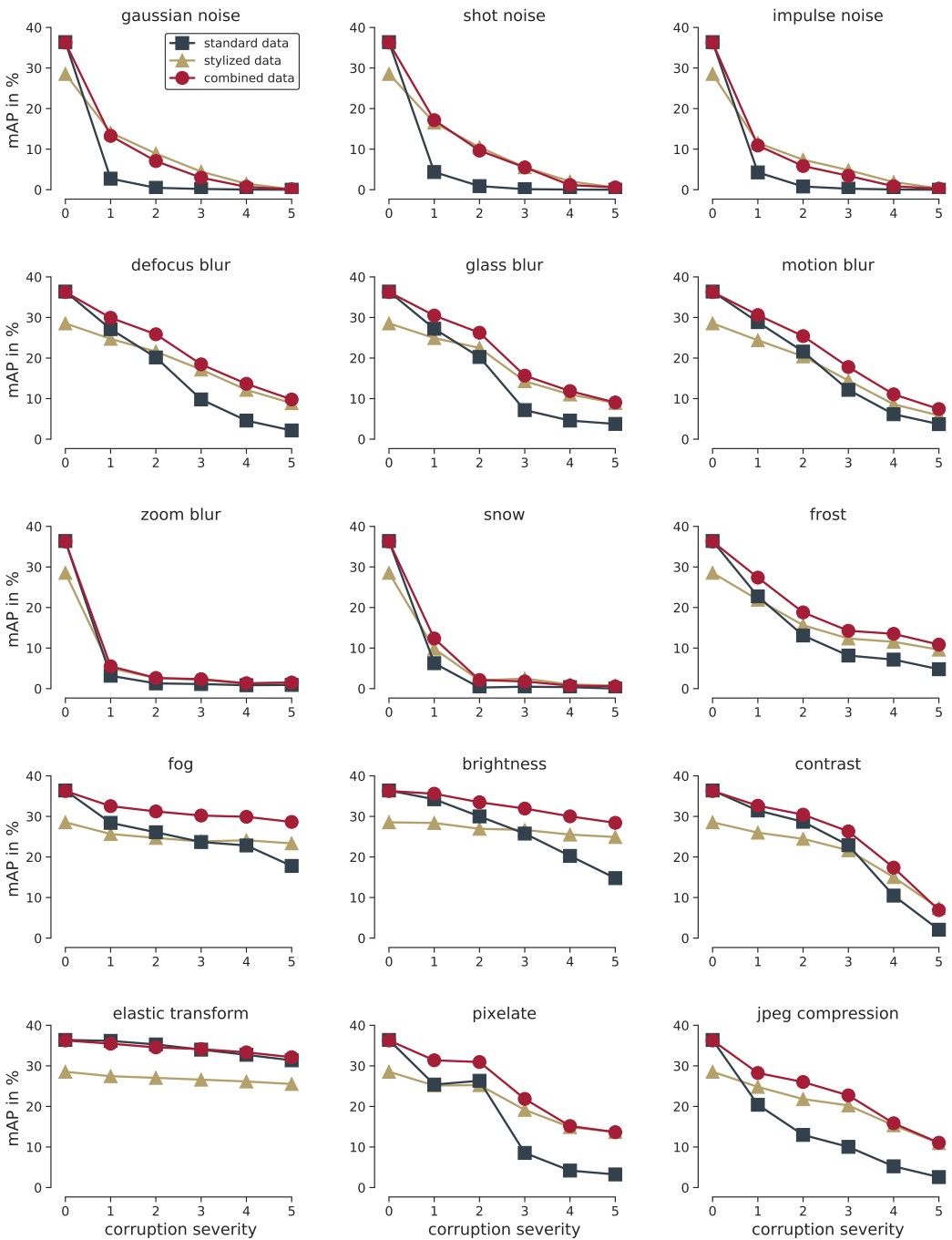

Figure 9: Results for each corruption type on Cityscapes-C.

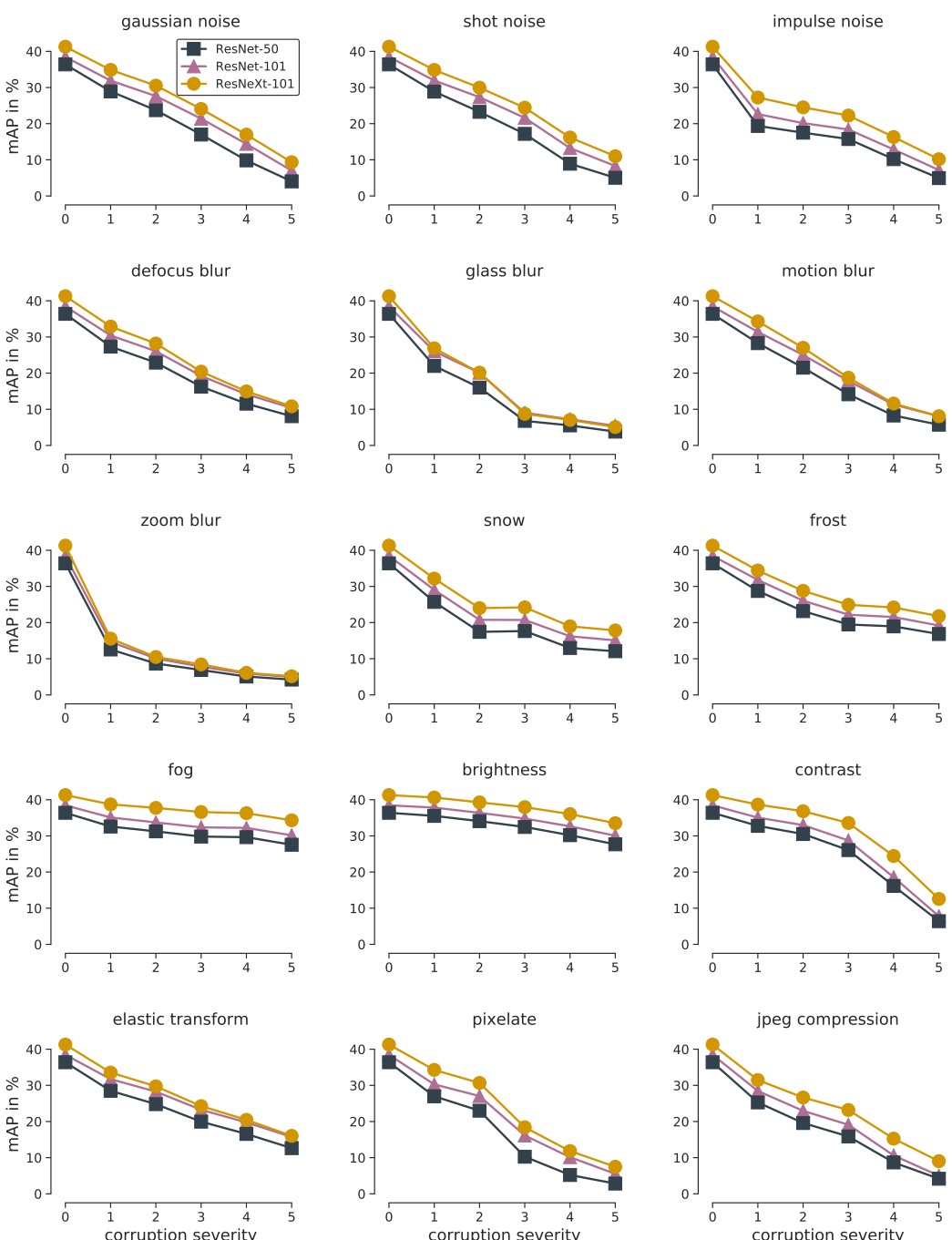

Figure 10: Results for each corruption type using different backbones. Faster R-CNN trained on MS COCO with ResNet-50, ResNet-101 and ResNext-101_64x4d backbones.

