# OpenReview forum: "Benchmarking Robustness in Object Detection: Autonomous Driving when Winter is Coming"
_ICLR.cc/2020/Conference — Reject_

### Official Review · AnonReviewer3 · 2019-10-19
**Official Blind Review #3**

**Rating:** 3

**Review:**

Summary:
- key problem or question: assessing / improving the robustness of object detectors to image corruptions (simulated fog, frost, snow, dragonfire...);
- contributions: 1) a benchmark (obtained by adding image-level corruptions to PASCAL, COCO, and Cityscapes) and an experimental protocol to measure detection robustness , 2) extensive experiments quantifying the severe lack of robustness of multiple state-of-the-art models, 3) experiments showing that data augmentation via style transfer (Geirhos et al, ICLR'19) improves robustness at little cost (at most -2% performance degradation on clean COCO images).

Recommendation: Weak Reject

Key reason: unclear novelty w.r.t. Geirhos et al ICLR 2019, especially due to the lack of specificity to object detection (which is the main goal of the paper).
- Although the paper from Geirhos et al. study a specific question (texture bias in CNNs), they propose a similar experimental protocol for assessing robustness, where the only differences are i) they use different corruptions (cf. Fig.6 in their paper, vs. Figs. 5, 7, 8, 9 here), ii) they do not have Cityscapes results (but they have PASCAL and COCO results).
- Furthermore, Geirhos et al 2019 also study the benefits of style-transfer-based data augmentation (this submission uses their technique), they report similar results and conclusions to this submission, including for object detection on PASCAL and COCO (cf. Table 2 in Geirhos et al 2019).
- What is, in the authors' opinion, the main differentiator of this submission? Does the difference in corruptions and evaluated models combined with the addition of Cityscapes yield new insights compared to Geirhos et al 2019?
- Beyond this similarity, what is specific to object detection vs. image classification in this submission? Besides the summary evaluation metrics, the corruptions and stylization are global (image-level) and not object-specific. Is the observed lack of robustness due to localization errors, mis-classifications, or other types of detection mistakes (cf. Hoiem et al ECCV'12)? What are the conclusions about detection robustness that differ from the image classification ones? What would be local corruptions that specifically degrade object detection performance (as studied for instance in the adversarial attack community, including physical attacks like Eykholt et al 2018)?

Additional feedback / questions:
- The results in section 3.4 / Fig.6 seem counterintuitive: more corruption should yield more degradation (as evidenced in Fig. 5 w.r.t. corruption severity). Is RMSE the right metric? Maybe SSIM would be better (more related to perceptual quality)?
- The aforementioned remark also raises the point that corruption difficulty night not be related to its intensity / noise level, but more with the (hard to quantify) domain gap w.r.t. clean data. For instance, if some images contain natural fog, fog corruption robustness should be naturally higher, whereas robust to never seen unrealistic dragonfire is expected to be naturally low. Can this relation between corruption and domain gap be somehow assessed? For instance using perceptual distance (using features from intermediate layers) to nearest clean neighbors? Or maybe by correlating with a subjective measure of realism of the corruption assessed globally per corruption type via a human study?
- Are corruption degradations dataset-specific? Fig. 7,8,9 seem to show different behaviors.
- How does the proposed benchmark compare to the Robust Vision Challenge (http://www.robustvision.net) proposed at CVPR 2018? How does robustness to corruptions and robustness across datasets correlate? Are they both similar "out of domain" robustness measures? How does this "out of domain" issue relate to adversarial examples (besides being "less extreme")?
- Could Fig. 5 include error bars / variance across corruption types?
- rPC is a good metric to compare models but I am not sure it is great to compare robustification methods, because when improving "clean" performance you mechanically decrease rPC, hence why combined is worse than stylized (e.g., in Table 2). What would be a better metric?
- Why does the stylized approach on COCO yields worse mPC (cf. Table 2 and 4), whereas it is expected to improve robustness under corruption (and does on other datasets), esp. since it sacrifices performance on the clean images by a lot?
- What is the impact of the choice of style images on robustness induced by stylization? Is diversity the most important factor (this can be tested by reducing the number of styles available)? If not, what is and how can it be measured?
- Do certain corruption (at certain severity levels) result in unrecoverable objects? For instance, dragonfire might completely occlude certain objects on the side, which might be a problem (e.g., there are frequent parked cars on the side in Cityscapes). What is the upper bound after corruption and how can it be measured?
- What is the human robustness for the new corruptions not present in Geirhos et al 2019? (Also relates to the aforementioned upper bound.)
- The paper is well written and I enjoyed the multiple pop culture references to Game of Thrones.

**Experience Assessment:**

I have published in this field for several years.

**Review Assessment: Checking Correctness Of Derivations And Theory:**

N/A

**Review Assessment: Checking Correctness Of Experiments:**

I carefully checked the experiments.

**Review Assessment: Thoroughness In Paper Reading:**

I read the paper thoroughly.

---

> ### Author Response · Authors · 2019-11-11
> **Clarification with respect to Geirhos et al. 2019**
>
> Thank you for your detailed review and excellent feedback! Before addressing your suggestions please allow us to quickly clarify one important point, which we think may be based on a misunderstanding: The unclear difference to Geirhos et al. (ICLR 2019) which you describe as the key reason for your score. We definitely need to make the difference more clear, thank you for pointing this out!
>
> Geirhos et al. indeed performed object detection experiments on MS COCO and Pascal VOC, showing that stylized ImageNet data helps for detection performance on *clean* images (through a better-performing ResNet backbone).
>
> The authors however never evaluated the detection robustness of their method (on *corrupted* images). Geirhos et al. do not show corruption robustness results on Pascal / COCO or any other detection dataset. Our proposed benchmark is to the best of our knowledge the first evaluation of detection robustness towards a large set of corruptions.
>
> For comparison reasons, we reimplemented their models and find only marginal robustness improvements from stylized pre-training of the ResNet backbone (<3% rPC):
>
> Results on COCO:
> Pre-training:           P [AP]        mPC        rPC
> IN                            31.8         15.5         48.7
> SIN                          29.8         15.3         51.3
> IN+SIN                    31.1         16.0         51.4
> IN+SIN ft IN             32.3         16.2         50.1
>
> Results on Pascal VOC:
> Pre-training          P [AP50]     mPC        rPC
> IN                            78.9         45.7        57.4
> SIN                          75.1         48.2        63.6
> IN+SIN                    78.0         50.6        64.2
> IN+SIN ft IN             79.0         48.9        61.4
> (abbreviations as in Geirhos et al.)
>
> In summary, stylized ImageNet data for training a ResNet backbone helps not only on clean data (as reported by Geirhos et al.) but also for corruption robustness, albeit less than our approach:
>
> For comparison the results of our method on COCO:
> Training:        P [AP]        mPC        rPC
> COCO              36.3         18.2        50.2
> Stylized            21.5         14.1        65.6
> Combined        34.6         20.4         58.9
>
> and on Pascal VOC:
> Training:       P [AP50]     mPC        rPC
> VOC                 80.5        48.6         60.4
> Stylized            68.0        50.0         73.5
> Combined        80.4        56.2         69.9
>
> We will add this experiment to the final version of the paper and try to make the difference more clear.
>
> We will address some more of your suggestions later on but will need some time for the experiments and wanted to quickly address this point right away.
>
> Implementation details:
> For the reimplementation, we follow Geirhos 2019 who use a slightly different model (Faster R-CNN without FPN, 31.8% AP) which performs a bit worse than the model we used throughout our paper (Faster R-CNN with FPN, 36.3% AP).
> Please also note that we use the standard metrics (AP for COCO and AP50 for Pascal VOC) to be consistent with the rest of our paper. In contrast, Geirhos et al. report AP50 on COCO. The 31.8% AP of our baseline translate to 52.6% AP50 which is close to the 52.3% AP50 reported in Geirhos 2019.

---

> > ### Author Response · Authors · 2019-11-15
> > **Additional experiments measuring the effect of corruptions on SSIM and performance of our method on natural distortions**
> >
> > Again, thanks a lot for your ideas!
> >
> > Following your suggestion we performed an analysis similar to our previous RMSE analysis but instead using SSIM. We indeed find a better but still limited correspondence between SSIM and the impact of each corruption. We updated the draft to include these experiments.
> >
> > We hope that your central concern concerning novelty was resolved through the clarification in the previous comment. This does not change the fact that our method was extending [8] and [9], as noted by the other reviewers. We agree with this assessment but still consider our benchmark highly relevant as object detection systems have a broad range of outdoor applications where they are confronted with natural distortions like weather changes. To strengthen this connection and add novelty we investigated whether results on our benchmark are related to natural distortions. We find this to be the case for images with annotations for weather conditions as well as highly realistic synthetic fog. Furthermoreour approach of training on the combination of normal and stylized data helps with every natural distortion we tested. For details please see our corresponding comment to Reviewer #2.
> >
> > We updated the draft to include these results as well as an analysis of pre-trained backbones in Sections 3.5 and 3.4 respectively.
> >
> > [8] Robert Geirhos, Patricia Rubisch, Claudio Michaelis, Matthias Bethge, Felix A. Wichmann, Wieland Brendel: ImageNet-trained CNNs are biased towards texture; increasing shape bias improves accuracy and robustness, ICLR 2019
> > [9] Dan Hendrycks, Thomas Dietterich: Benchmarking Neural Network Robustness to Common Corruptions and Perturbations, ICLR 2019

---

### Official Review · AnonReviewer1 · 2019-10-20
**Official Blind Review #1**

**Rating:** 3

**Review:**

This paper introduces a  benchmark to assess the performance of object detection models when image quality degrades. Three variants of detection datasets, termed PASCAL-C, COCO-C and Cityscapes-C, are introduced that contain a large variety of image corruptions. The paper shows that standard object detection models suffer a severe performance loss on corrupted images (down to 30–60% of the original performance). Further, this work shows that a simple data augmentation trick of stylizing the training images leads to a substantial increase in robustness across corruption type, severity and dataset.

The paper is well written and easy to follow. The proposed benchmark is interesting and clearly show the deficiencies of state-of-the-art object detection methods in case of image corruptions or weather conditions. However, my main concern is the novelty in that the proposed approach is just an extension of [1]. [1] introduced corrupted versions of commonly used classification datasets (ImageNet-C, CIFAR10-C) as standardized benchmarks. The different types of corruptions used here for object detection and their sorting into four groups were also introduced originally in [1]. Moreover, the idea to use style transfer as an augmentation to  improve corruption robustness for image classification has been introduced in [2]. Therefore, the only contribution of this paper is to apply the ideas from [1, 2] for object detection.

[1] Dan Hendrycks and Thomas Dietterich. Benchmarking neural network robustness to common corruptions and perturbations. In ICLR, 2019.
[2] Robert Geirhos, Patricia Rubisch, Claudio Michaelis, Matthias Bethge, Felix A. Wichmann, and Wieland Brendel. ImageNet-trained CNNs are biased towards texture; increasing shape bias improves accuracy and robustness. In ICLR, 2019.

**Experience Assessment:**

I have published one or two papers in this area.

**Review Assessment: Checking Correctness Of Derivations And Theory:**

I assessed the sensibility of the derivations and theory.

**Review Assessment: Checking Correctness Of Experiments:**

I carefully checked the experiments.

**Review Assessment: Thoroughness In Paper Reading:**

I read the paper thoroughly.

---

> ### Author Response · Authors · 2019-11-15
> **New experiments to demonstrate the relevance of our benchmark and method for real world corruptions**
>
> Thank you for your feedback and your assessment of our article as being “interesting and clearly show[ing] the deficiencies of state-of-the-art object detection methods”.
>
> Your main concern is the novelty of our approach. We agree that our work builds on extending the works of [8] and [9] for the object detection community. It was previously unclear to which degree detection models would be affected by image distortions. While our result may not be entirely surprising - the behavior is very similar to that of object recognition models - we consider it even more relevant for the object detection community which often struggles with “natural” distribution shifts inherent to applying models “in the wild”.
>
> We therefore aimed to address your concern by performing additional experiments on such *natural* distortion shifts induced by real-world (rather than synthetic) corruptions such as different weather conditions and the transition from daytime to nighttime, answering the (previously open) question of whether robustness towards synthetic distortions also implies robustness on real-world distortions, which goes much further than [8] and [9]. Our findings suggest that performance on the proposed benchmarks is indeed predictive for performance on natural distribution shifts. Moreover, combined training on normal and stylized data improves performance in every single experiment (for heavy fog, performance improves even for about 50%. Details are included in Section 3.5 of the updated PDF). While collecting data for real-world distortions is often prohibitively expensive, this opens up a “cheap” and effective proxy evaluation method that may be of interest to all those interested in building robust detection methods for “the wild”.
>
> We hope that this analysis addresses your concern in two ways. It adds novelty because such an analysis of both synthetic and real-world distortions has to the best of our knowledge never been conducted, and it demonstrates the relevance of the proposed benchmark when addressing distribution shifts that occur naturally.
>
> [8] Robert Geirhos, Patricia Rubisch, Claudio Michaelis, Matthias Bethge, Felix A. Wichmann, Wieland Brendel: ImageNet-trained CNNs are biased towards texture; increasing shape bias improves accuracy and robustness, ICLR 2019
> [9] Dan Hendrycks, Thomas Dietterich: Benchmarking Neural Network Robustness to Common Corruptions and Perturbations, ICLR 2019

---

### Official Review · AnonReviewer2 · 2019-10-24
**Official Blind Review #2**

**Rating:** 3

**Review:**

This paper presents a benchmark for measuring robustness to input image corruption in object detection settings. The paper proposes a benchmark for this task, and proposes a simple data augmentation technique for this task.

Strengths
1. Understanding the robustness properties of existing vision models is an important problem.
2. The paper establishes a sensible protocol for the benchmark, where methods are tested upon image perturbations that are not used for training the model.
3. I like the proposed simple data augmentation procedure and the experimental finding that data augmentation with such a procedure leads to models that are robust to held-out, previously unseen perturbations.

Shortcomings:
1. While the paper proposes a sensible experimental protocol, certain questions remain:
a) Are the set of test time perturbations exhaustive and representative of the perturbations in the real world? The paper doesn't talk about this, or provides any experimental data to establish this. The paper derives them from an earlier paper called "Benchmarking neural network robustness to common corruptions and perturbations", and thus I am not even sure if the proposed set of perturbations should be viewed as a contribution of the current paper.
b) While the paper itself follows good practice by not training on perturbations that are considered at test time, unfortunately, it does not define a clear protocol or characterization as to how future researchers should use the benchmark. I believe setting up such a protocol is going to be difficult and is worthy of more thought and consideration, absence of this weakens the paper, as it leaves the door open for flawed future research.

2. Missing comparisons: Proposed method is interesting, but I wonder if there were a more standard evaluation to test the efficiency of the method, perhaps something like testing if representations learned using such data augmentations were more robust to adversarial perturbations? Or perhaps, comparison against other methods that exist in literature for related tasks, such as methods that study how to make networks robust to adversarial perturbations?

Because of the aforementioned reasons, I don't view the benchmarking part of the paper as a solid contribution. Similarly, the proposed method is simple and intuitive (which is good), but it will help if there were more comparisons to set the paper in context of related work.

**Experience Assessment:**

I do not know much about this area.

**Review Assessment: Checking Correctness Of Derivations And Theory:**

N/A

**Review Assessment: Checking Correctness Of Experiments:**

I assessed the sensibility of the experiments.

**Review Assessment: Thoroughness In Paper Reading:**

I made a quick assessment of this paper.

---

> ### Author Response · Authors · 2019-11-11
> **Relationship between corruption robustness and adversarial robustness**
>
> There have been several recent publications showing that robustness towards common corruptions (as studied here) and robustness towards adversarial perturbations are not related. Engstrom et al. report that increasing robustness against adversarial L_infinity attacks does not increase robustness against translations and rotations [4]. Jordan et al. show that adversarial robustness does not transfer easily between attack classes [5]. Most importantly, Laugros et al experimentally show that adversarial robustness and robustness towards common corruptions are independent [6], a finding which is corroborated by Anonymous et al. [7]. Hence, there is no strong link between corruption and adversarial robustness, and we will add a paragraph to the related work section of our paper to point future readers to this insight.
>
>
> [4] Logan Engstrom, Dimitris Tsipras, Ludwig Schmidt, Aleksander Madry: A rotation and a translation suffice: Fooling CNNs with simple transformations.
> [5] Matt Jordan, Naren Manoj, Surbhi Goel, Alexandros G Dimakis: Quantifying perceptual distortion of adversarial examples.
> [6] Alfred Laugros, Alice Caplier, Matthieu Ospici: Are Adversarial Robustness and Common Perturbation Robustness Independent Attributes?
> [7] Anonymous: When Robustness Doesn’t Promote Robustness: Synthetic vs. Natural Distribution Shifts on ImageNet (ICLR 2020 submission)

---

> ### Author Response · Authors · 2019-11-11
> **Additional experiments to demonstrate the relevance to corruptions in the real world**
>
> Thank you for your valuable feedback and for your assessment of our work as addressing “an important problem”. We provide a point-by-point response to your suggestions below.
>
> 1a. Connection to perturbations in the real world
>
> This is a good point, we assumed that improved performance on synthetic distortions automatically leads to better performance on perturbations in the real world—but we never actually showed this in the paper. Following your suggestion, we conducted two experiments to investigate this link:
>
> I) We evaluated our approach (training on a combination of stylized and normal training data) on Foggy Cityscapes [1], a version of Cityscapes with realistic fog. In the presence of strong fog, a vanilla detection model shows strong performance impairments (down to 51% of its original performance). Our model, on the other hand, is far more robust on this real-world distortion (retaining 72% of original performance).
>
> Absolute (AP) clean    0.005     0.01      0.02  (number indicates strength of fog)
> Standard         36.4       30.2      25.1     18.7
> Ours* 		36.3       32.2      29.9     26.2
>
> Relative (%)	clean    0.005     0.01      0.02
> Standard	100       83.0      69.0      51.4
> Ours*		100       88.7      82.4      72.2
>
> *  model trained jointly on stylized and clean data, denoted “combined” in paper
>
> II) We measure the effect of our method on other natural distortions using weather annotations from the BDD100k dataset [2], which contains real-world rain and show images. The BDD100k distortions are relatively benign (e.g., images annotated as rainy often show only wet roads but no real rain). Consequently, the performance drop of a standard model is small. Nevertheless, our approach outperforms the baseline model on both “rainy” and “snowy” conditions.
>
> BDD100k:
> Weather:	Clear       Rainy      Snowy
> Standard	27.8          27.6         23.6
> Ours*		27.7          28.0         24.2
>
> Taken together, these two additional experiments conducted in response to your suggestion establish a link between synthetic corruptions (as employed by our benchmark) and natural distortions (real-world rain, snow, fog) as evaluated on the Foggy Cityscapes / BDD100k datasets. Interestingly, good performance on synthetic corruptions is predictive for performance on natural distortions, which is a novel finding that has, to the best of our knowledge, never been reported before. As data for real-world distortions is often prohibitively expensive to collect, our benchmark opens up an efficient proxy method for assessing the effect of natural distortions.
>
> 1b. Protocol for benchmark use by future researchers
>
> We agree: defining a clear protocol on how to use the benchmark in the future is key. Our previous description of the submission process (how to submit, which metrics to report, how models will be ranked, …) was buried in the Appendix, which probably is not salient enough to ensure a consistent evaluation protocol for future users of our benchmark. We will thus add a parapraph to section 2.1 (“Robust Detection Benchmark”).
>
>
> We would appreciate it if you could take the time to assess these new results and indicate whether this addresses your suggestions and changes your assessment of our work. Please feel free to point out anything else you might consider relevant.
>
>
> [1] Christos Sakaridis, Dengxin Dai, and Luc Van Gool: Semantic Foggy Scene Understanding with Synthetic Data, International Journal of Computer Vision (IJCV), 2018
> [2] Fisher Yu, Wenqi Xian, Yingying Chen, Fangchen Liu, Mike Liao, Vashisht Madhavan and Trevor Darrell: BDD100K: A Diverse Driving Video Database with Scalable Annotation Tooling, Arxiv 2018
> [3] Agrim Gupta, Piotr Dollár and Ross Girshick: LVIS: A Dataset for Large Vocabulary Instance Segmentation, Arxiv 2019

---

> > ### Author Response · Authors · 2019-11-15
> > **Submission updated**
> >
> > We updated the article to include the experiments and results on natural distortions.

---

### Decision · Program_Chairs · 2019-12-19

**Decision:**

Reject

**Comment:**

This paper proposes a benchmark for assessing the impact of image quality degradation (e.g. simulated fog, snow, frost) on the performance of object detection models. The authors introduce corrupted versions of popular object detection datasets, namely PASCAL-C, COCO-C and Cityscapes-C, and an evaluation protocol which reveals that the current models are not robust to such corruptions (losing as much as 60% of the performance). The authors then show that a simple data augmentation scheme significantly improves robustness. The reviewers agree that the manuscript is well written and that the proposed benchmark reveals major drawbacks of current detection models. However, two critical issues with the paper paper remain, namely lack of novelty in light of Geirhos et al., and how to actually use this benchmark in practice. I will hence recommend the rejection of this paper in the current state. Nevertheless, we encourage the authors to address the raised shortcomings (the new experiments reported in the rebuttal are a good starting point).